# Use of Stromal Intervention and Exogenous Neoantigen Vaccination to Boost Pancreatic Cancer Chemo-Immunotherapy by Nanocarriers

**DOI:** 10.3390/bioengineering10101205

**Published:** 2023-10-16

**Authors:** Saborni Chattopadhyay, Yu-Pei Liao, Xiang Wang, André E. Nel

**Affiliations:** 1California NanoSystems Institute, University of California, Los Angeles, CA 90095, USA; 2Division of NanoMedicine, Department of Medicine, David Geffen School of Medicine, University of California, Los Angeles, CA 90095, USA; 3Jonsson Comprehensive Cancer Center, University of California, Los Angeles, CA 90095, USA

**Keywords:** pancreatic cancer, tumor stroma, irinotecan, lipoxins, KRAS vaccination, tertiary lymphoid structures

## Abstract

Despite the formidable treatment challenges of pancreatic ductal adenocarcinoma (PDAC), considerable progress has been made in improving drug delivery via pioneering nanocarriers. These innovations are geared towards overcoming the obstacles presented by dysplastic stroma and fostering anti-PDAC immune reactions. We are currently conducting research aimed at enhancing chemotherapy to stimulate anti-tumor immunity by inducing immunogenic cell death (ICD). This is accomplished using lipid bilayer-coated nanocarriers, which enable the attainment of synergistic results. Noteworthy examples include liposomes and lipid-coated mesoporous silica nanoparticles known as “silicasomes”. These nanocarriers facilitate remote chemotherapy loading, as well as the seamless integration of immunomodulators into the lipid bilayer. In this communication, we elucidate innovative ways for further improving chemo-immunotherapy. The first is the development of a liposome platform engineered by the remote loading of irinotecan while incorporating a pro-resolving lipoxin in the lipid bilayer. This carrier interfered in stromal collagen deposition, as well as boosting the irinotecan-induced ICD response. The second approach was to synthesize polymer nanoparticles for the delivery of mutated KRAS peptides in conjunction with a TLR7/8 agonist. The dual delivery vaccine particle boosted the generation of antigen-specific cytotoxic T-cells that are recruited to lymphoid structures at the cancer site, with a view to strengthening the endogenous vaccination response achieved by chemo-immunotherapy.

## 1. Introductory Statement

Pancreatic ductal adenocarcinoma (PDAC) represents a significant challenge in the field of oncology, ranked as the twelfth most common cancer worldwide, with 495,000 new cases in 2020 [1]. In the United States, it is the fourth leading cause of cancer-related deaths, where patients diagnosed with PDAC face an alarming median survival of less than 6 months, and the five-year survival rate remains distressingly in the single-digit range [2]. One of the most challenging aspects of PDAC is its tendency to be detected at an advanced stage, often limiting the feasibility of surgical intervention, which is crucial for successful treatment in many other cancer types. Consequently, patients are frequently left with limited treatment options. Despite extensive research efforts, the response to traditional chemotherapy or immunotherapy, including the use of checkpoint-blocking antibodies, has been disappointing. Over the years, we have introduced several nanotechnology-enabled strategies for the treatment of PDAC aimed at improving drug delivery by nanocarriers that provide improved pharmacokinetics and the use of single or synergistic drug combinations that trigger cytotoxic cell death and/or immunogenic cell death (ICD) for chemo-immunotherapy strategies. The purpose of this communication is to discuss new groundbreaking studies recently undertaken as we continue to develop nano-enabled chemo-immunotherapy in PDAC. After a brief overview of our long-term research strategy, we discuss new cutting-edge experiments that address the use of a stroma-targeting nanocarrier delivering lipoxin A4 (LXA4), as well as the use of exogenous KRAS vaccination to strengthen the chemotherapy-induced immune response.

## 2. Nano-Enabled Chemo-Immunotherapy in PDAC

Pancreatic ductal adenocarcinoma (PDAC) poses a formidable challenge due to late diagnosis, limited surgical accessibility, and chemotherapy resistance. Moreover, PDAC displays a robust dysplastic stroma that hinders drug delivery and contributes to a tumor microenvironment (TME) that impedes the immune response. Nanocarriers offer a promising solution to some of these impediments, including improved drug delivery and providing chemo-immunotherapy [3,4,5]. One historical approach has been the introduction of an albumin-bound paclitaxel nanocarrier (Abraxane) [6], acting to reduce stromal density while also allowing improved gemcitabine uptake at the tumor site. Building on this, we developed a lipid-coated mesoporous silica nanoparticle (MSNP) carrier that could improve similar drug synergy by allowing the contemporaneous release of gemcitabine and paclitaxel [7]. This was made possible by the remote loading of gemcitabine into the porous interior of the carrier with paclitaxel incorporation into the lipid bilayer. This carrier is also referred to as a silicasome. Another nano-based approach was the development of a liposomal carrier (Onivyde) [8] for irinotecan, one of four drugs in the FOLFIRINOX regimen [9], which are collectively more effective than gemcitabine but highly toxic [10]. Onivyde showed improved efficacy in combination with 5-fluorouracil and leucovorin in the phase III NAPOLI-1 trial in metastatic, gemcitabine-resistant PDAC patients; the formulation had leakage issues and residual toxicity, resulting in an FDA black box warning [11]. To address these challenges, we once more resorted to the use of a silicasome to improve irinotecan efficacy by using the supported lipid bilayer that provides improved drug loading with more stable drug retention (Figure 1A) [12,13].

In addition to improving irinotecan delivery via the silicasome and an orthotopic PDAC tumor model, we demonstrated that this topoisomerase inhibitor initiates an immunogenic cell death (ICD) response (Figure 1A) [13]. ICD leads to the release of endogenous tumor antigens from dying cancer cells, which express calreticulin on the cell surface to provide an “eat-me” signal to antigen-presenting cells (APC). Moreover, ICD is accompanied by the release of adjuvant stimuli (ATP and HMGB1) that promote the activation and maturation of dendritic cells in the TME. This amounts to the equivalent of an endogenous vaccination response. Complementing the ICD effect, irinotecan also produces a cell stress response, resulting in PD-L1 expression; this allowed us to demonstrate synergistic boosting of the irinotecan-induced ICD response by anti-PD1 monoclonal antibodies in an orthotopic KRAS tumor model [13,16]. A similar outcome could also be achieved by silicasome carriers and liposomes delivering other ICD-inducing chemotherapeutic agents, e.g., an oxaliplatin derivative (DCH-Pt), also boosted by anti-PD1 [17]. These results illustrate the promise of immune checkpoint inhibitors (ICI) to boost the ICD impact in PDAC, a disease often accompanied by low neoantigen burden, limiting the frequency of T-cell infiltrates to 20–30% nontreated human tumors [18,19,20]. It is also worth commenting that even with the achievement of synergy between irinotecan chemo-immunotherapy and anti-PD1 administration, orthotopic KRAS tumors often present with heterogeneous immune landscapes, requiring additional therapeutic intervention, including dealing with a dysplastic stroma and an immune-suppressed tumor immune microenvironment (TIME) [4].

Given the requirement of additional therapeutic combinations for effective chemo-immunotherapy, we further adapted silicasome and liposome design to allow the lipid bilayer to be used for combination therapy, premised on drug loading into the aqueous interior, as well as in the lipid bilayer (Figure 1A) [4]. This design strategy is premised on remote loading of ICD-inducing chemotherapeutics (e.g., irinotecan, doxorubicin, and mitoxantrone), using the lipid bilayer for the encapsulation of protonating agents (e.g., ammonium sulfate, citric acid, and sucrose octa sulfate), allowing the chemo drugs to cross the bilayer. The lipid bilayer also allows the incorporation of hydrophobic drugs (e.g., paclitaxel and 3M-052) or lipid-conjugated drugs (e.g., prodrugs attached to cholesterol or phospholipids), in addition to the remote-loaded chemo agents [4,12]. As an example, we synthesized liposomes and silicasomes that can co-deliver irinotecan with a TLR7 agonist, 3M-052, anchored into the lipid bilayer via a lipid tail (Figure 1B) [12]. This design allowed the boosting of the irinotecan-induced ICD response in an orthotopic PDAC model via the ability of TLR7 to enhance dendritic cell activation and T-cell recruitment to the PDAC site. By so doing, 3M-052 could overcome the paucity of poorly activated APC at the cancer site, strengthening the cancer immunity cycle (Figure 1C) [12]. This combination strategy is currently being extended by including additional immunomodulators and checkpoint blocking antibodies for synergistic immunotherapy [4].

Another variation of the theme shown in Figure 1A was to combine the ICD-inducing chemo agent, mitoxantrone, with a cholesterol-conjugated inhibitor of the indoleamine-pyrrole 2,3-dioxygenase (IDO-1) pathway, indoximod, in bilayer carriers for the treatment of triple-negative breast cancer, colon, and lung cancer [21]. In addition to the demonstration of drug synergy, the ICD response to mitoxantrone included the interesting observation that this particular drug also boosted tumor immune surveillance by NK cells [21]. The idea of drug conjugation to bilayer components was further developed by making use of medicinal chemistry criteria to introduce additional prodrug design strategies for small molecule inhibitors of PD-1, glycogen, synthase kinase-3 (GSK3), adenosine A2 receptor (A2AR), and chemokine C-X-C receptor 4 (CXCR4) [4]. These strategies can also be combined with additional remote loading strategies for silicasomes and liposomes to deliver GSK3 and CXCR4 inhibitors, with additional impacts on the diversity of immune landscapes that appear in PDAC, triple-negative breast cancer, colon, and lung cancer. GSK3 impacts the PD1/PD-L1 axis by transcriptional suppression of PD1 expression, leading to the interference of immune escape with similar efficacy as anti-PD1 antibodies, while the inhibition of CXCR4 was effective in restoring T-cell exclusion from the tumor core heterogeneous triple negative breast cancer in PDAC landscapes [4,14].

A further important area to consider for PDAC immunotherapy is the development of nanocarriers to target the desmoplastic stroma, other than using the paclitaxel/gemcitabine silicasome discussed earlier [4,7]. Important non-transformed cellular targets in the stroma include cancer-associated fibroblasts (CAFs), tumor-associated macrophages, lymphocytes, myeloid-derived suppressor cells (MDSCs), and granulocytes. Table 1 lists a number of current therapeutic options for stromal targeting, including cytokine and chemokine pathways that facilitate tumor invasion and spread [22,23]. We are particularly interested in the role CAFs play in programming the TME by a range of soluble mediators, including transforming growth factor-beta (TGF-β), interleukin 6 (IL-6), matrix metalloproteinases (MMPs), vascular endothelial growth factor (VEGF), hepatocyte growth factor (HGF), stromal cell-derived factor 1 (SDF-1), hypoxia-inducible factor-1 (HIF-1), and tissue inhibitors of metalloproteinases (TIMPs) [24]. CAFs are also involved in cancer drug resistance via the modulation of WNT signaling. TGF-β activation is particularly significant in paracrine signaling by CAFs and mediating TME responses such as extracellular matrix (ECM) deposition, epithelial–mesenchymal transition (EMT), angiogenesis, and tumor metastasis [25]. The CAF secretome further plays a role in the recruitment of tumor-associated macrophages (TAMs), which contributes to the development of an immunosuppressive TME [26,27,28]. Table 1 illustrates some of the drug options for targeting CAFs in pre-clinical studies. These conform to three categories: (i) the depletion of ECM proteins, (ii) inhibiting ECM protein synthesis, and (iii) the blockade of signaling pathways leading to CAF activation and stromal deposition. Among these, our immediate interest is in lipoxins that provide anti-inflammatory “stop signals” engaged in leukocyte trafficking and infiltration of the TME [29]. As such, lipoxins can reduce chemo-induced inflammation in the TME by stimulating the removal of tumor debris by macrophages and lowering drug requirements [29,30]. From a nanotherapeutic perspective, pro-resolving lipid mediators exhibit short circulatory half-lives and short residence time at sites of tissue inflammation, making them ideal candidates for encapsulation in our lipid-based carriers for stromal therapy, as will be discussed in the following subsection.

The second innovation we will address pertains to our goal of strengthening the endogenous vaccination response to chemo-immunotherapy by combining that approach with an exogenous vaccination strategy that intersects in the cancer immunity cycle (Figure 1C) [29,30]. We already commented on the low neoantigen burden in PDAC as a key contributor resulting in poorly immunogenic immune landscapes, in contradiction to melanoma or non-small cell lung cancer. While triggering of an immune “hot-start” by ICD-inducing chemotherapeutics improves the immunogenicity of the landscape, the supply of endogenous tumor-associated antigens (TAAs) (e.g., carcinoembryonic antigen and mesothelin) or neoantigens (e.g., KRAS and p53) may not suffice to educate high-affinity cytotoxic T lymphocytes (CTL) recruited to the primary cancer site from regional lymphoid structures. This concept is embedded in the cancer immunity cycle (Figure 1C) [15]. Our hypothesis is that exogenous vaccination with frequently expressed TAA or neoantigens may boost the availability of high-affinity CTL clones to participate in the ICD-triggered immune cycle (Figure 1C). Accordingly, the final section of this communication will discuss the use of encapsulated mutant KRAS peptides to strengthen PDAC immunity.

### 2.1. Development of a Dual-Delivery Irinotecan/Lipoxin A4 Liposome to Target the Stroma and Improve PDAC Immunotherapy

The desmoplastic PDAC stroma creates a barrier that limits drug delivery to the TME, thereby hampering the effectiveness of conventional therapies. In addition, the stroma creates an immunosuppressive environment, making it difficult for the immune system to effectively target and eliminate cancer cells [23]. Finding ways to reduce desmoplasia is considered a promising approach to improving PDAC therapy. Arachidonic acid-derived eicosanoids produced via the CYP450, cyclo-oxygenase (COX), or lipoxygenase (LOX) enzymatic pathways are key factors in carcinogenesis [48]. This family of bioactive lipids includes leukotrienes, hydroxy-eicosatetraenoic acids, lipoxins, and resolvins, which regulate several pathophysiological processes in the body, including inflammation, cellular proliferation, angiogenesis, vascular flow, extracellular matrix deposition, and immune function [49]. Of particular significance to the stroma and TME in PDAC is the role of leukotrienes and lipoxin/resolvins in the initiation and resolution of inflammatory processes. While acute inflammation is generally protective against injurious stimuli, uncontrolled chronic inflammation encourages carcinogenesis by DNA injury, epigenetic dysregulation, genomic instability, and/or changes in intracellular signaling [48]. Several experimental studies have shown the benefits of LOX-derived lipoxins and resolvins in suppressing tumorigenesis and adverse effects on immunity in the setting of chronic inflammation [50,51]. Our continuing efforts are focusing on the role of lipoxin A4, based on the demonstration that a lipoxin score shows a good correlation to human PDAC metastatic potential [52]. Moreover, cellular and animal studies using the enzymatically stable LXA4 analog BML-111 were shown to interfere with the invasive capacity of pancreatic cancer cells [30], as well as the differentiation of human pancreatic stellate cells into a CAF-like phenotype [39]. This is of particular importance to our PDAC treatment objectives, where chemotherapy can induce refractory inflammation that impacts the dysplastic stroma and the recruitment of immunosuppressive cells.

Despite these promising leads, the full clinical potential of pro-resolving lipid mediators is still to be realized because of rapid degradation in the circulation and at sites of local tissue inflammation. LXA4, for instance, is rapidly metabolized by human monocytes via dehydrogenation and a reduction to 13,14-dihydro LXA4 [53]. To overcome this challenge, more biodurable analogs have been developed to prevent metabolic degradation and to prolong therapeutic effects, e.g., BML-111, discussed earlier [53]. Based on its enzymatic instability, we propose developing a liposome to improve LXA4 delivery in combination with irinotecan (Figure 2A,B). This was accomplished using ammonium sulfate as a protonating agent for irinotecan remote loading while incorporating lipophilic LXA4 into the lipid bilayer (Figure 2A and Appendix A). Liposome synthesis was accomplished by a thin film hydration technique during which LXA4 was dissolved in the organic phase, with ammonium sulfate added to the hydration phase. Subsequent remote loading was accomplished by incubating the vesicles in a buffer containing dissolved irinotecan, allowing drug import by the generation of a proton gradient across the lipid bilayer (Figure 2A). Dynamic light scattering was used to assess the size (80–90 nm) and zeta potential (−3 mV) of the liposomes (Table 2). CryoEM confirmed the presence of the irinotecan drug precipitate within the aqueous core of the liposomes (Figure 2C). The functionality of the liposomes was confirmed in vitro, using human pancreatic stellate cells to show interference in IL-6 production in the presence of TGF-β (data not shown). This reflects the anti-inflammatory effect of lipoxin on the TGF-β signaling pathway by binding to the high-affinity G protein-coupled lipoxin A4 (LXA4) receptor and formyl peptide receptor 2 (FPR2)/ALX at nanomolar quantities [29,30].

An animal study was performed in B6/129J mice to determine the impact of the liposomes on the subcutaneous growth of a KRAS transformed murine pancreatic adenocarcinoma (KPC) cell line harvested from a spontaneously developing tumor in a transgenic KrasLSL G12D/+; Trp53LSL R172H/+; Pdx-1-Cre mouse [54]. The mice were intravenously injected with the LXA4-irinotecan liposomes, delivering 90 μg/kg LXA4 plus 40 mg/kg irinotecan twice a week for 2 weeks (Appendix A). Assessment of the tumor volumes demonstrated that animals treated with the LXA4-irinotecan liposomes had the slowest growth rate compared to irinotecan monotherapy or saline control (Appendix A). Following animal sacrifice and tumor harvesting, Masson’s trichrome staining was performed to determine the impact on TME collagen density (Figure 3A). This demonstrated that in animals treated with the dual-drug (LXA4/irinotecan) liposome, there was a significant reduction in collagen staining intensity compared to other groups, including for the liposome delivering irinotecan only (Figure 3A). The staining intensity was quantitatively confirmed by ImageScope software analysis (Figure 3B). Harvested tumor tissues were subsequently used for multicolor immunohistochemistry (IHC) to assess the relative abundance of CD8^+^ CTLs vs. FoxP3^+^ regulatory T-cells (T_regs_) in the TME. This demonstrated while encapsulated irinotecan as well, the dual delivery (LXA4/irinotecan) liposome delivery led to a reduction in T_reg_ expression (Appendix A,B), the most significant increase in CD8^+^/FoxP3^+^ ratio was achieved by the dual delivery liposome (Figure 3C). Additional IHC staining for perforin expression confirmed increased cytolytic activity during treatment with the dual-delivery liposome compared to other treatment groups (Figure 3D and Appendix A). Noteworthy, the immunostimulatory effects of the LXA4-irinotecan combination could proceed without evidence of animal toxicity (e.g., weight, hematology, liver, and kidney function).

In summary, the combination of LXA4 with irinotecan in a liposomal carrier was effective for synergistic interference in collagen deposition and immune cell infiltration in a subcutaneous KPC model. Similar experiments are planned for a silicasome carrier in an orthotopic KPC model to determine the effect on a robust dysplastic stroma that resembles human PDAC. These studies will include a detailed investigation into the reprogramming of immune suppressive cellular infiltrates in the TME in PDAC, e.g., switching M2 TAMs to an M1 phenotype [55]. We will also consider incorporating additional drugs undergoing clinical trials (Table 1) to study their comparative efficacy and will also combine LXA4 in liposome and silicasome carriers that co-deliver immunomodulators that target the CXCR4, IDO-1, and TLR7 pathways.

### 2.2. Combined Use of Mutant KRAS Vaccination Approach to Boost the Endogenous Vaccination Response to ICD in PDAC

The aim of this section is to delineate the use of polymeric nanoparticles for generating a vaccine response to the mutant KRAS_G12D_ epitope to boost chemo-immunotherapy. The potential advantage of such an approach would be to enhance the recruitment of high affinity, antigen-specific CTL clones from lymphoid organs (e.g., lymph nodes and spleen) or tertiary lymphoid structures that may participate in the cancer immunity cycle (Figure 1C). The hypothesis is that low levels of endogenous TAA and neoantigens that are being released from dying cancer cells during ICD may not be effective for stimulating T-cell antigen receptors, leading to the recruitment of CTLs that exhibit low cytotoxic potential for cancer cells. Thus, exogenous vaccination with PDAC antigens, known to lead to effective MHC-I presentation, could populate the regional lymphoid structures with higher affinity CTL populations for recruitment to the primary cancer site. In addition to the spleen and regional lymph nodes, there is now growing awareness of the role of tertiary lymphoid structures (TLS), assembling at primary cancer sites for boosting antitumor immunity [56,57,58]. TLS are organized structures similar to lymphoid follicles that harbor T-cells, B-cells, and dendritic cells, with the ability to improve immune recruitment, activation, proliferation, and cytotoxic killing. Accordingly, the appearance of TLS in various cancers is regarded as predictive of improved of antitumor immune responses and patient outcomes, including for pancreatic cancer [59]. Thus, we hypothesize that exogenous vaccination will improve the impact of endogenous vaccination in PDAC by ICD.

KRAS frequently presents driver mutations in PDAC, colorectal, and lung cancers, with mutation frequencies of 97.7, 44.7, and 30.9%, respectively [60]. Four mutations (G12, G13, Q61, G12D) cover more than 99% of the PDAC mutations, with KRAS G12D contributing up to 45% (Figure 4) [61,62,63]. Not only do these PDAC mutations drive the development of pancreatic intraepithelial neoplasia but prediction making for selecting immunogenic KRAS peptides to perform vaccination has also greatly improved. This includes evidence that KRAS G12D can be expressed on type I murine MHC and human HLA (e.g., HLA-A*11:01 or HLA-C*8:02) antigen-presenting complexes, with the ability to induce PDAC targeting by CTL [64,65]. This has resulted in demonstrating therapeutic responses to mutated KRAS peptides in human PDAC trials [66,67]. Thus, while some phase I and II clinical trials have shown longer survival in vaccine responders compared to non-responders [68], no therapeutic benefit was obtained in advanced-stage pancreatic cancer patients. We further propose that vaccine efficacy can be improved in combination with ICD therapy, thereby allowing higher affinity CTL to participate in the cancer immunity cycle (Figure 1C). From a vaccination perspective, it is also relevant that the administration of an irradiated allogeneic PDAC vaccine (GVAX) could boost the generation of tertiary lymphoid structures in a human PDAC trial [69,70].

Since the immunogenicity of peptide vaccines is enhanced by lymph node (LN) delivery, we surmised that the same approach could be applicable to the generation of TLS in our murine model [71,72,73]. Typically, the uptake of vaccinating nanoparticles in LNs is facilitated by a particle size of 25 to 200 nm. It was also demonstrated that the immunogenicity of peptide vaccines can be enhanced by conjugating the peptides to TLR agonists [74] or a carrier protein such as albumin [75,76,77]. The strategy that we adopted was to use passive KRAS peptide encapsulation into poly(lactic-co-glycolic acid) (PLGA) nanoparticles that also include a TLR7/8 agonist, resiquimod (R848) (Figure 4 and Figure 5). PLGA nanoparticles are avidly taken up by LNs, allowing efficient encapsulation and cargo release for antigen presentation in LNs. Carrier synthesis was achieved by using a double-emulsion solvent evaporation technique to construct two distinct nanoparticle versions (Figure 5A). In the first embodiment, the unmodified M23 peptide, sparingly soluble in water, was added to the oil phase containing PLGA in dichloromethane, with R848 being added in the entrapped aqueous phase. This yielded particles that encapsulate the M23 peptide in the shell, with R848 in the aqueous core. In the second embodiment, the KRAS G12D peptide was modified on both ends with aspartate residues to enhance water solubility (Figure 4). This peptide was added to the inner aqueous phase, combined with R848, while the oil phase was comprised of PLGA solubilized in DCM (Figure 5). Using the double emulsion approach, another nanoparticle was constructed where both the hydrophilic D23 peptide and R848 were encapsulated in the aqueous core (Figure 5). We also synthesized an M23/DiR nanoparticle with an empty core and a scrambled peptide (polyD) nanoparticle encapsulating the peptide with R848 in the aqueous core (Figure 5C). This yielded nanoparticles in the 150–200 nm size range with a zeta potential of −70 mV, conducive to lymphatic spread. The SEM images further confirmed the size and polydispersity distribution (Figure 5B). Both D23 and M23 containing nanoparticles showed a 23% loading efficiency of R848.

To demonstrate the agonist activity of the encapsulated R848, the nanoparticles were co-incubated with murine HEK-Blue cells to demonstrate a dose-dependent increase in TLR7-mediated reporter gene activity (data not shown). We also employed the DiR-labelled nanoparticle to demonstrate its lymphatic spread to regional lymph nodes when injected subcutaneously in B6/129J mice (not shown). Compared to the free DiR, the particle-encapsulated dye was retained for a prolonged period during live in vivo imaging (IVIS).

To determine vaccination impact on anti-tumor immunity, B6/129J mice were subcutaneously injected with (M23 + R848) _PLGA_ and (D23 + R848)_PLGA_ nanoparticles 13 days prior to subcutaneous challenge with KRAS pancreatic cancer cells (KPC). As a control for the impact of the nanoparticles, we also used subcutaneous injection of KPC cells treated with an ICD-inducing dose (300 mM) of irinotecan to demonstrate interference in live KPC growth. In one flank of the animal, 5 × 10^4^ KPC cells undergoing ICD were administered to induce a protective immune response, demonstrated by the growth failure of live KPC cells injected in the contralateral flank a few days later [13].

The possible protective effect of the nanoparticles was tested by injecting the animals in one flank on three occasions, seven days apart, with particles delivering peptide doses of 10 μg M23 or D23 plus 2 μg R848. Additional controls included animal groups vaccinated with an admix of free peptide plus adjuvant (i.e., M23 + R848 admix and D23 + R848 admix). A day prior to the final boost, the mice were challenged with 1.5 × 10^5^ live KPC cells subcutaneously on the contralateral flank and observed for tumor growth, as described in Figure 6A. The animals were monitored for 45 days, during which tumor volumes and animal weights were recorded (Figure 6B,C). This demonstrated that 50% of mice vaccinated with (M23 + R848)_PLGA_ remained tumor-free for up to 45 days after the challenge (Figure 6B). Furthermore, in tumor-developing animals, there was significantly slower growth in the nanoparticle-treated animals compared to unvaccinated or animals receiving the free (M23 + R848) admix. All the animals vaccinated with irinotecan-treated KPC cells remained tumor-free, indicating the efficacy of the ICD effect. In contrast to the results obtained with the encapsulated (M23 + R848)_PLGA_ combination, animals vaccinated with (D23 + R848)_PLGA_ had higher instances of tumor formation, where 5 out of 7 vaccinated mice (70%) developed tumors. However, like (M23 + R848)_PLGA_ vaccination, the animals receiving D23 vaccine nanoparticles showed considerable slowing of tumor growth compared to admix vaccinated mice (Figure 6C).

The remaining alive mice were sacrificed 45 days after the challenge for tumor harvesting and IHC analysis. IHC staining to demonstrate the presence of CD8^+^ tumor-infiltrating lymphocytes confirmed a significant increase in CTLs in tumors from mice vaccinated with both (M23 + R848)_PLGA_ and (D23 + R848)_PLGA_ nanoparticles compared to the free admix or unvaccinated mice (Figure 6D,E). Representative IHC panels are shown in Appendix A. We also observed the emergence of tertiary lymphoid structures (TLS), a feature of tumor immunity that has been more frequently recognized recently [78,79]. Noteworthy, examination of the tumors in (M23 + R848)_PLGA_ vaccinated mice demonstrated the appearance of intra-tumoral lymphoid aggregates in 50% of tumors, with an illustrative example shown in (Figure 6F). This is illustrated by the H&E staining, as well as the IHC analysis of CD8^+^ CTLs and CD21^+^ follicular dendritic cells in these structures (Figure 6F). In contrast to the results with the M23 encapsulated peptide, only 20% of tumors in the (D23 + R848)_PLGA_ vaccine group developed lymphoid aggregates. In summary, the (M23 + R848)_PLGA_ vaccine elicited a stronger and more persistent immune activation compared to (D23 + R848)_PLGA_, evidenced by fewer instances of tumors, smaller tumor sizes, and the consistent formation of lymphoid aggregates.

To further assess the efficacy of the KRAS vaccine in an orthotopic KPC-luciferase model, we selected the more robust (M23 + R848)_PLGA_ nanoparticle for prophylactic administration and histological analysis. Using the same vaccination timeline and dosimetry as in Figure 6A, tumor-bearing mice were vaccinated with the encapsulated (M23 + R848)_PLGA_ combination (Figure 7A) prior to the implantation of 0.5 × 10^6^ KPC-luciferase cells in the pancreas, a day prior to the final boost. The control groups included animals that were not vaccinated or vaccinated with either M23 nanoparticles (without R848) or nanoparticles delivering a scrambled peptide (polyD) plus R848. The mice were monitored every 4 to 5 days for tumor progression using IVIS imaging (Figure 7B,C). Mice were sacrificed 16 days post-challenge, followed by harvesting of tumors and other key organs for ex vivo imaging. This demonstrated the presence of smaller tumors with reduced luminescence intensity in mice immunized with (M23 + R848)_PLGA_, compared to unvaccinated or animals receiving the (polyD + R848)_PLGA_ or M23_PLGA_ (no adjuvant) vaccines (Figure 7D). The subsequent performance of ex vivo IVIS analysis also demonstrated larger tumor masses with metastatic spread to surrounding organs in non-treated animals, while animals receiving (M23 + R848)_PLGA_ showed significantly smaller tumors, judged by tumor luminescence and weight (Figure 7E,F). This response was also significantly improved over animals vaccinated with (polyD + R848)_PLGA_ or M23_PLGA_. The observations suggest that control mice receiving injections with peptide or adjuvant showed an inferior response compared to animals receiving nanoparticles encapsulating both M23 and R848.

Further demonstration of the immunogenicity of the (M23 + R848)_PLGA_ vaccine was obtained by performing ELISpot assays on splenocytes retrieved from vaccinated and unvaccinated mice, followed by ex vivo stimulation with the corresponding KRAS peptide. This demonstrated an increased number of IFNγ^+^ and TNFα^+^ producing cellular spots in mice vaccinated with (M23 + R848)_PLGA_ nanoparticles (Figure 8A,B). In comparison, unvaccinated animals or animals treated with M23_PLGA_ showed fewer ELISpots. Interestingly, there was an increase in the number of IFNγ^+^ colonies in the spleens of mice receiving (polyD + R848)_PLGA_, which is of uncertain significance considering that the same does not happen for TNFα. Tumor sections were also analyzed for T-cell infiltration, demonstrating an increased number of CD8^+^ and CD4^+^ T lymphocytes in the (M23 + R848)_PLGA_ vaccination group (Figure 8C,D). The corresponding IHC panels appear in Appendix A. Moreover, like the results in the subcutaneous experiment, all the (M23 + R848)_PLGA_ vaccinated mice showed the development of lymphoid aggregates (Figure 8E). These immune clusters included positive staining for CD4, CD8, and CD21 cellular markers, which is a characteristic of TLS. Despite their potential prognostic relevance, the actual mechanism involved in TLS generation at primary cancer sites and the precise contribution to antitumor immunity is incompletely understood [80]. One possibility is that these structures provide an intra-tumoral reservoir of immune cells, available for antigen sampling and potential CTL recruitment to the tumor cores. This hypothesis is now being tested in the orthotopic model, where KRAS vaccination is combined with chemo-immunotherapy, as well as other immunomodulators.

## 3. Conclusions and Future Prospects

Cancer’s intricate relationship with peripheral tolerance and imperfect adaptive response is well established. The efficacy of treatment varies according to the tumor mutational burden, which is often low enough that even with the addition of checkpoint blocking antibodies, only 20–30% of tumors respond [18,19,20]. Notably, pancreatic tumors exhibit heightened ICI resistance due to the development of an immune-suppressive TME. Within this context, our communication introduces two innovative approaches: stroma-ablative chemo-immunotherapy and the application of exogenous KRAS vaccine nanoparticles. These strategies offer distinct mechanisms that hold promising translational prospects. They have the potential to transform aggressive, stroma-rich, and immune-evading tumors into entities that can be recognized by the immune system.

A major future prospect will be to combine nano-enabled chemo-immunotherapy with exogenous vaccination provided by nanocarriers that deliver PDAC-specific neo-antigens and tumor-associated antigens (TAAs). We hypothesize that this will lead to the strengthening of the cancer immunity cycle, as depicted in Figure 1C. The exogenous vaccination strategy could benefit from a variety of additional smart design features, including a selection of multiple mutant KRAS epitopes that can be assembled into a single vaccine to cover the heterogeneous tumor landscape that may include more than one KRAS mutant. This vaccine will also be valuable for the treatment of cancers other than PDAC. Moreover, neoantigen epitopes can be combined with TAAs (e.g., mucin 1 and mesothelin) that, although less immunogenic, are more stably expressed. A multi-epitope strategy will increase the number of antigen-specific T-cell clones that can be recruited from adjacent lymphoid structures (including TLS) [69,79] to participate in the polyclonal immune response triggered by ICD-inducing chemotherapeutic agents. One of the approaches for facilitating multi-epitope delivery includes the use of nucleic acid analogs that can be linked in a single mRNA strand that can be delivered by cationic lipid nanoparticles. These epitope-delivering nanocarriers can also be endowed with surface ligands that target lymphoid structures, such as the mannose receptor, widely expressed in antigen-presenting cells in lymph nodes and the spleen. In addition, the particles could be endowed with a TLR7 agonist that also serves to strengthen antigen presentation and immune recruitment in lymphoid tissue, which could possibly be extended to TLS developing at the primary cancer site. Thus, an array of vaccination carriers can be developed for use in combination with the liposomes and silicasomes that deliver chemo-immunotherapy agents, plus a number of co-packaged immunomodulators, as described by us [4].

## Figures and Tables

**Figure 1 bioengineering-10-01205-f001:**
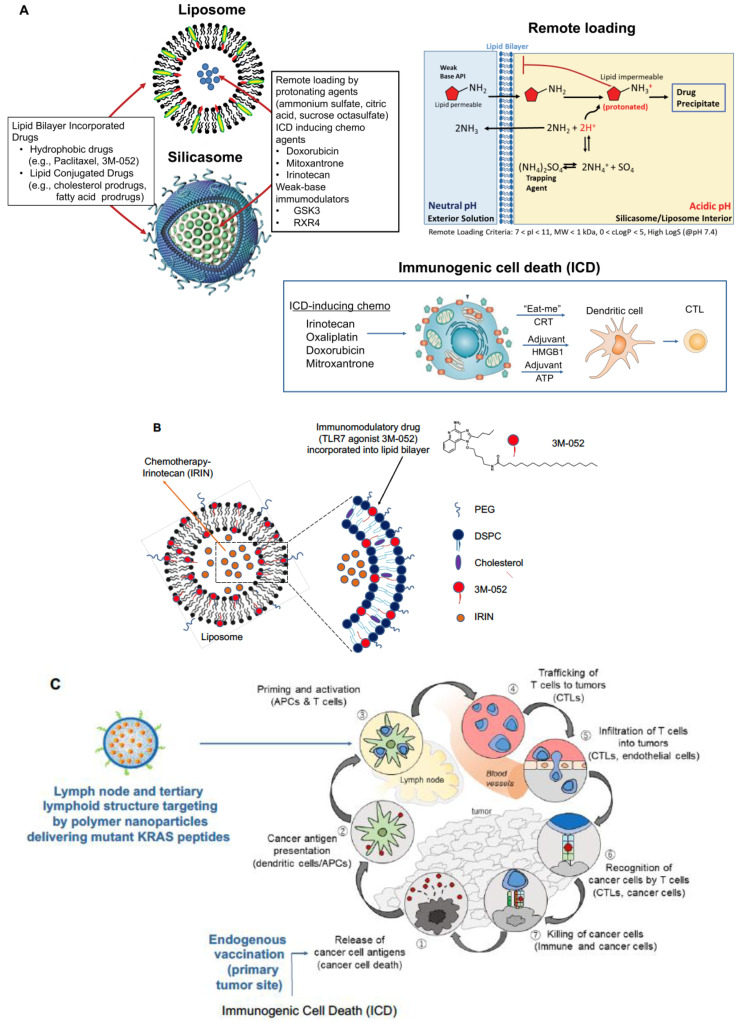
**Use of lipid bilayer encapsulated nanocarriers for endogenous chemo-immunotherapy, to be boosted by KRAS neoantigen-delivering polymer nanoparticles**. (**A**). Schematic to illustrate combination drug loading into liposomes and silicasomes, including remote loading of chemotherapeutic agents that induce immunogenic cell death (ICD), such as irinotecan, doxorubicin, and mitoxantrone. The schematic also explains the use of protonating agents (e.g., ammonium sulfate, citric acid, and sucrose octa sulfate) for remote import across the lipid bilayer, also serving as a carrier component for incorporating hydrophobic drugs (e.g., paclitaxel and 3M-052) or lipid-conjugated prodrugs, capable of synergizing with the chemotherapeutic agents. The bottom diagram illustrates the ICD response pathway, which depends on dying cancer cells displaying calreticulin (CRT) expression, acting as an “eat me” signal for antigen-presenting cells, which also receive adjuvant stimuli (high mobility group box 1 or HMGB 1, and ATP) to promote the maturation of T-cell activating dendritic cells. All considered, the ICD response acts as a response pathway releasing endogenous tumor antigens participating in the cancer immunity cycle. Reprinted from Biomaterials, Volume 269, Allen et al., “Immune checkpoint inhibition in syngeneic mouse cancer models by a silicasome nanocarrier delivering a GSK3 inhibitor”, © 2021 with permission from Elsevier [14]. (**B**) Illustrative example of a liposome co-delivering irinotecan with a hydrophobic TLR7 agonist, 3M-052. The induction of ICD by irinotecan is augmented by incorporating the lipid tail of the TLR7 agonist into the lipid bilayer. The release of this agonist enhances the recruitment and activation of dendritic cells participating in the ICD response. (**C**) Schematic illustrating the cancer immunity cycle [15] explaining the hypothesis of using exogenous vaccination with KRAS peptides to strengthen the endogenous vaccination response by increasing the frequency of antigen-specific cytotoxic T-cells (CTLs) to be recruited from regional lymph nodes, spleen, and tertiary lymphoid structures for tumor cell killing at the primary cancer site. The illustration of the immunity cycle was adapted from Immunity, Volume 39 (1), Daniel S. Chen and Ira Mellman, Oncology Meets Immunology: The Cancer-Immunity Cycle, Pages 1–10, © 2013 with permission from Elsevier.

**Figure 2 bioengineering-10-01205-f002:**
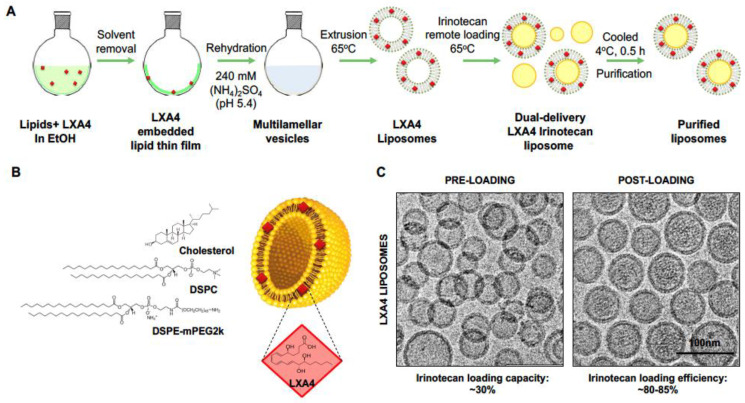
**Synthesis of LXA4 embedded irinotecan liposomes:** (**A**) Schematic showing the synthetic steps to construct a dual delivery liposome by a thin film hydration method, which includes the use of ammonium sulfate for subsequent irinotecan remote loading. (**B**) An illustration of the structure of the LXA4-irinotecan liposomes, the bilayer composition of which is described in detail in Appendix A. (**C**) Images of the resulting liposomes were obtained with a cryo-electron microscope (TF20 FEI TecnaiG2) that demonstrated uniform liposomes with a clear unilamellar, bilayer structure and large aqueous core. After remote loading, the presence of drug precipitates was visible by a darker contrast of the liposomal interior. Scale = 100 nm.

**Figure 3 bioengineering-10-01205-f003:**
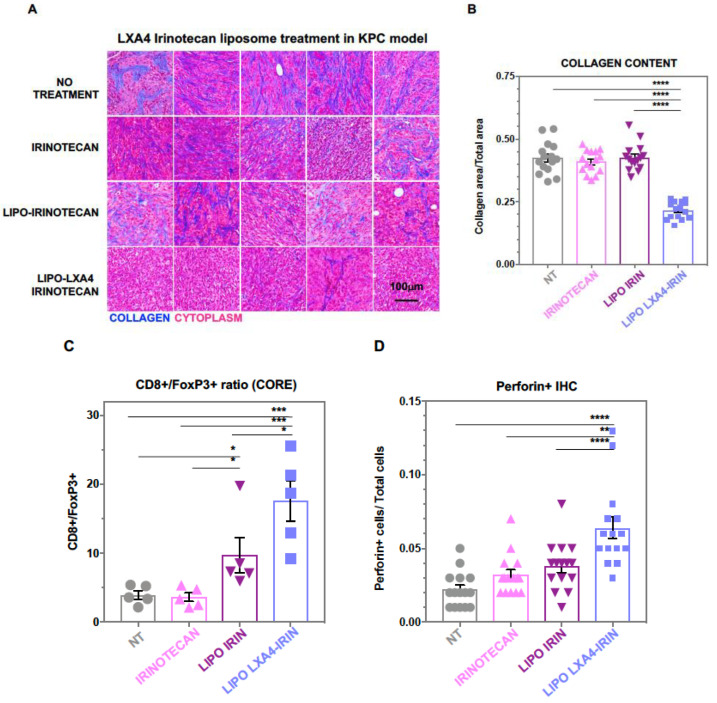
**In vivo therapeutic efficacy of liposomal LXA4-irinotecan in a subcutaneous KPC pancreatic cancer model.** Mice with palpable tumors were injected with either free or liposomal irinotecan (40 mg/kg) or LXA4 irinotecan (90 μg/kg LXA4; 40 mg/kg irinotecan) twice a week (for a total of 4 injections) starting at day 6 post-implantation. After 21 days of tumor implantation, animals were sacrificed under anesthesia; tumors were harvested and fixed in 4% formalin, followed by paraffin embedding and sectioning into 5 μm sections. (**A**,**B**) Tumor sections were analyzed and quantified for collagen content by Masson’s trichrome staining with ImageScope software. (**C**) Tumor-infiltrating T-lymphocytes were analyzed and quantified as the ratio of cytotoxic (CD8^+^) to (FoxP3^+^) regulatory T lymphocytes by inForm software, used, to analyze multicolor IHC slides as described in Appendix A. (**D**) Cytotoxic T-cell activity was analyzed by perforin IHC and quantified with ImageScope software (Appendix A). Data represent mean ± SEM. The images were analyzed for statistical significance by one-way ANOVA. Differences were considered significant for a *p*-value of * *p* < 0.05, ** *p* < 0.01, *** *p* < 0.001, **** *p* < 0.0001, respectively. Scale bar = 100 μm.

**Figure 4 bioengineering-10-01205-f004:**
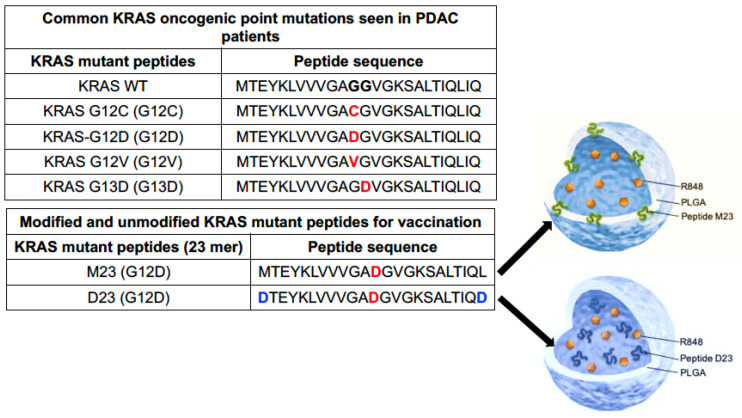
**KRAS mutations involved in PDAC generation and peptide selection for constructing vaccinating nanoparticles.** The upper table outlines common KRAS point mutations found in PDAC patients, from which we selected the G12D mutation for constructing mutant peptides for nanoparticle synthesis. Owing to the hydrophobicity of the native mutant 23 mer KRAS peptide, termed M23, we modification of its terminal amino acids on either end with aspartate residues to enhance its solubility, creating a hydrophilic 23 mer termed D23. The illustration further shows how the mutant KRAS peptides (M23 or D23) are incorporated into the polymeric shell and aqueous core, respectively, along with the TLR7/8 agonist, R848. This was accomplished by a modified double emulsion solvent evaporation method. Briefly, a mixture of KRAS G12D mutated peptides (23 mer) and R848 were added to PLGA, dissolved in dichloromethane (DCM), and sonicated to create a water-in-oil emulsion (w1/o). This emulsion is added to a 1% solution of sodium cholate and sonicated again to form a water in oil in water double emulsion (w1/o/w2). Vaccine particles were harvested after evaporating solvent overnight under continuous stirring at room temperature in 0.1% sodium cholate solution by ultracentrifugation. Red indicates the point mutation and blue the addition of amino acids to changes solubility index.

**Figure 5 bioengineering-10-01205-f005:**
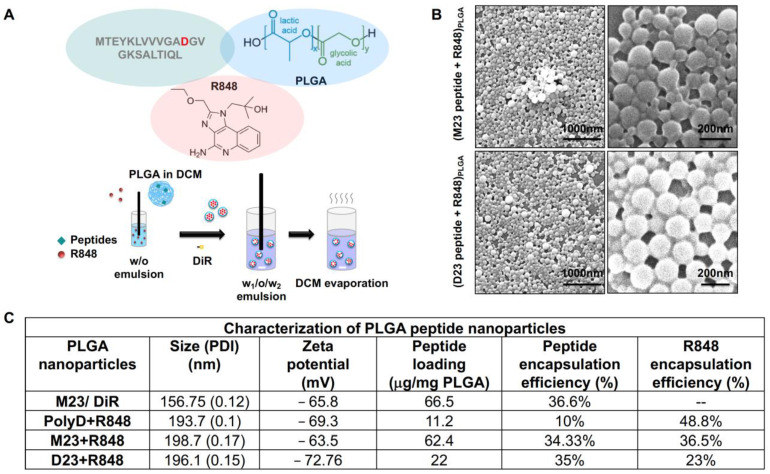
**Synthesis of polymeric nanoparticle vaccines with hydrophobic mutated KRAS peptides:** (**A**) Schematic showing the modified double emulsion solvent evaporation method to generate hollow nanoparticles with M23 peptide (G12D point mutation highlighted in red) embedded in the polymeric shell and passive encapsulation of R848 in the aqueous core. The particles are harvested in 0.1% sodium cholate solution by ultracentrifugation before analyzing for size and zeta potential with dynamic light scattering (DLS). (**B**) Images of the vaccine nanoparticles were obtained with SEM that demonstrated largely monodisperse nanoparticles with a smooth surface. Scale bar = 200 nm. (**C**) Table describing the DLS characterization of the various vaccine nanoparticles and control peptide nanoparticles that were synthesized for the animal studies.

**Figure 6 bioengineering-10-01205-f006:**
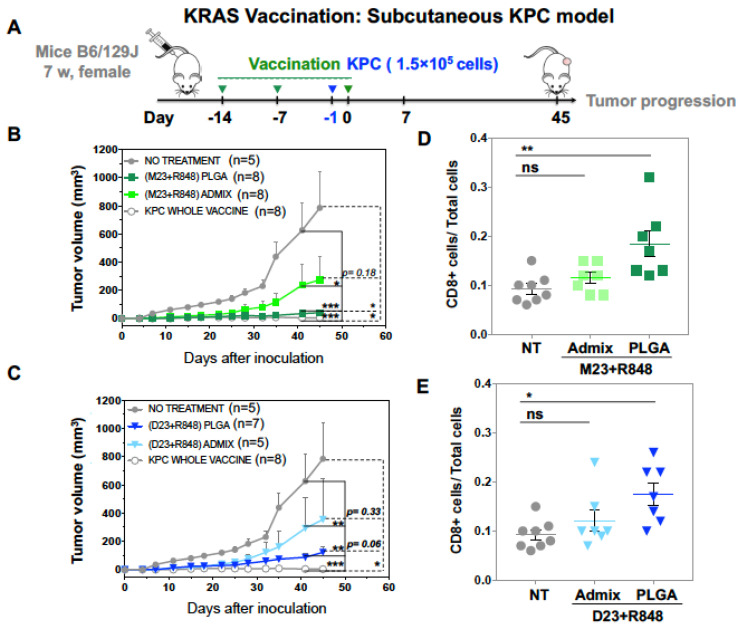
**In vivo prophylactic efficacy of the KRAS vaccine nanoparticles in a subcutaneous KPC pancreatic cancer model.** All animal experimental protocols were approved by the UCLA Animal Research Committee. (**A**) For the subcutaneous KRAS vaccine model, female B6129SF1/J mice (7 weeks old) were vaccinated subcutaneously on the right hind flank and boosted twice every 7 days with 10 μg peptide in nanoparticulate or free form as described in the schematic. Control groups included mice vaccinated with irinotecan-treated KPC cells for mice not receiving vaccination. The animals were challenged subcutaneously on the left hind flank with 0.1 × 10^6^ viable KPC cells (in a 1:1 mixture of PBS and Matrigel) a day before the final boost. (**B**,**C**) Mean tumor volumes of the challenged mice across the treatment groups were recorded for the duration of the studies. Following 45 days of tumor challenge, animals were sacrificed under anesthesia; tumors were harvested and fixed in 4% formalin followed by paraffin embedding and sectioning into 5 μm sections. (**D**,**E**) The tumor sections were analyzed for tumor-infiltrating T lymphocytes by IHC and quantified for CD8^+^ staining intensity by ImageScope software. (**F**) H&E and IHC images showing intratumoral lymphoid aggregates. The table describes the frequency of tumors in the different treatment groups in the fraction of tumors developing lymphoid aggregates. Data represent mean ± SEM. Statistical analysis was performed by one-way ANOVA. Differences were considered significant for a *p*-value of * *p* < 0.05, ** *p* < 0.01, *** *p* < 0.001 respectively and not significant (ns) for *p* > 0.05. Scale bar = 200 μm.

**Figure 7 bioengineering-10-01205-f007:**
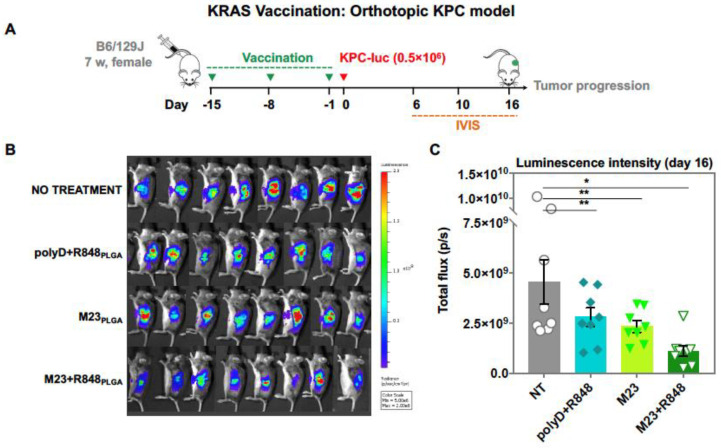
**In vivo prophylactic efficacy of the KRAS vaccine nanoparticles in an orthotopic KPC-luciferase pancreatic cancer model**. (**A**) For the orthotopic KRAS vaccine model, female B6129SF1/J mice were vaccinated subcutaneously on the right hind flank and boosted twice, seven days apart, with 10 μg peptide (M23 + R848)_PLGA_ compared to unvaccinated or animals receiving the (polyD + R848)_PLGA_ or M23_PLGA_ (no adjuvant) vaccines. Thirteen days following the commencement of the vaccination as described in the schematic, 0.5 × 10^6^ viable KPC luciferase cells (<12 passages) suspended in a 1:1 PBS to Matrigel solution were directly injected into the pancreas for orthotopic tumor growth. During the study, the mice were monitored for tumor growth by (**B**) recording total flux of in vivo luciferin signal from each mouse and (**C**) quantifying prior to sacrifice. (**D**) Mice were terminated on day 16 post-tumor implantation and analyzed for metastatic spread with ex vivo imaging of the luciferin signal from (**E**) tumors and internal organs in the peritoneal cavity. (**F**) The weights of the harvested tumors were also recorded. Data represent mean ± SEM, n = 8. Statistical analysis was performed by one-way ANOVA. Differences were considered significant for a *p*-value of * *p* < 0.05, ** *p* < 0.01 respectively and not significant (ns) for *p* > 0.05.

**Figure 8 bioengineering-10-01205-f008:**
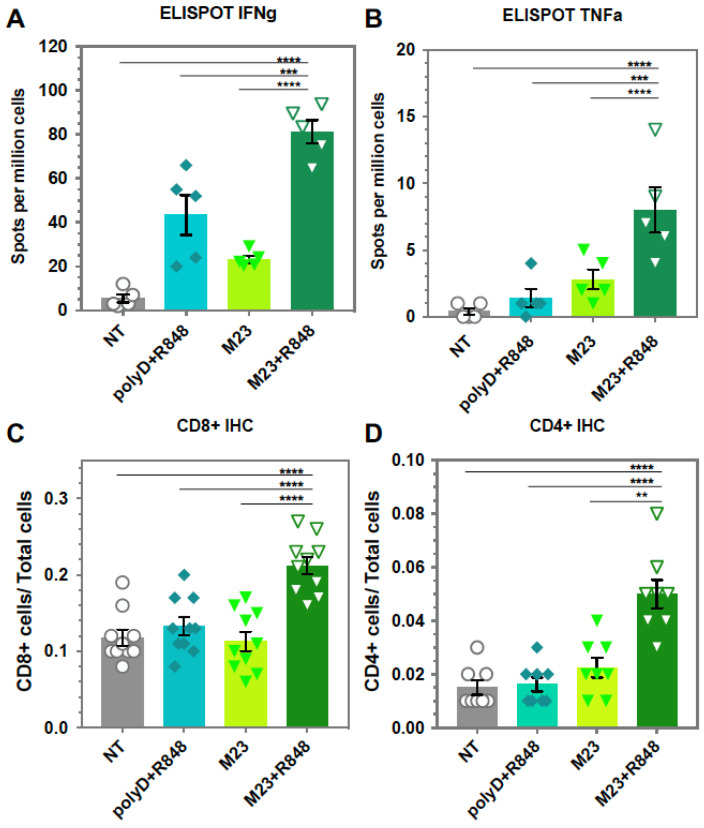
**Assessment of the immune landscape in orthotopic KPC mice, following vaccination with peptide nanoparticles.** Animals used for the experimentation in Figure 7 were sacrificed on day 16 after initial tumor implantation. Splenocytes from vaccinated mice were harvested for ex vivo stimulation and conducted ELISpot assays that assess (**A**) INFγ and (**B**) TNFα release. Primary tumors were harvested, fixed, embedded, and sectioned into 5 μm sections. These sections were analyzed by IHC and quantified for (**C**) CD8^+^ and (**D**) CD4^+^ tumor-infiltrating T lymphocytes using ImageScope software. (**E**) IHC staining for CD8, CD4, and CD21 to characterize the appearance of intra-tumoral TLS in (M23 + R848)_PLGA_ vaccinated mice. The table describes the fractions of tumors in each treatment developing TLS. Data represent mean ± SEM, n = 8. Statistical analysis was performed by one-way ANOVA. Differences were considered significant for a *p*-value of ** *p* < 0.01, *** *p* < 0.001, **** *p* < 0.0001, respectively. Scale bar = 500 μm.

**Table 1 bioengineering-10-01205-t001:** Drug targeting of CAFs and their mechanism of action in PDAC.

Target	Active Agent	Pre-Clinical Model	Mechanism	References
ECM	PEGPH20	Pancreatic tumor	Depletion of hyaluronic acid	[31,32]
ECM	Pamrevlumab (FG-3019)	Pancreatic tumor	Reduction in Connective Tissue Growth Factor (CTGF) expression	[33,34]
ECM	Simtuzumab (GS-6624)	Pancreatic tumor	Lysyl oxidase inhibitor	[35]
ECM	Losartan	Pancreatic tumor	Angiotensin inhibition reduces stromal collagen and hyaluronan production	[36]
CAF-cancer cell crosstalk	Resveratrol	Pancreatic tumor	Stromal remodeling by reducing the number of CAFs and leukocytes	[37]
ECM	Captopril	Pancreatic tumor	TGF-β pathway inhibition	[38]
ECM	Lipoxin A4	Pancreatic tumor	TGF-β pathway inhibition	[39]
CAF-cancer cell crosstalk	Fraxinellone	Pancreatic tumor	TGF-β pathway inhibition	[40]
CAF-cancer cell crosstalk	Triptolide	Pancreatic tumor	TGF-β pathway inhibition	[41]
ECM	Cyclopamine	Pancreatic tumor	Hedgehog pathway inhibition	[42,43,44]
CAF activation	Vismodegib (GDC-0449)	Pancreatic tumor	Hedgehog pathway inhibition	[45]
CAF-cancer cell crosstalk	Curcumin	Pancreatic tumor	EMT inhibition	[46]
ECM	Pirfenidone	Pancreatic tumor	MMP-2 reduction	[47]

**Abbreviations:** CAF = cancer-associated fibroblasts; ECM = extracellular matrix; EMT = endothelial-mesenchymal transition.

**Table 2 bioengineering-10-01205-t002:** Characterization of LXA4 liposomes.

Liposomes (0.2 mol% LXA4)	Size (PDI)	Zeta Potential	Loading Capacity	Loading Efficiency
(Units)	(nm)	(mV)	(%)	(%)
Pre-loading	88.2 (0.15)	−3.06 ± 2.1	--	--
Post loading	76.2 (0.09)	−2.53 ± 3.87	28.95	82.09

## Data Availability

The data presented in this study are available upon request from the corresponding author.

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
