# Peer review of "Use of Stromal Intervention and Exogenous Neoantigen Vaccination to Boost Pancreatic Cancer Chemo-Immunotherapy by Nanocarriers"

_bioengineering, 2023, doi:10.3390/bioengineering10101205_

Round 1
Reviewer 1 Report
Excellent research work
The conclusions made are fully confirmed by experimental data.
The authors have provided a scheme of the experiment, which makes it easier to understand.
It is also worth noting that the quality of statistical processing and data visualization is very high.
The work can be accepted for publication in this form
Author Response
We thank the reviewer for a diligent review and laudatory responses (excellent work; conclusions confirmed by experimental data; easy to understand; high quality of statistical data), in addition to agreeing that the work is acceptable for publication.
Reviewer 2 Report
This communication is interesting and probably provides good insight to readership.
Author Response
We thank the reviewer for a diligent review and laudatory responses that the paper is interesting and provides good insight into the covered topic.
Reviewer 3 Report
Dear Authors;
Re: bioengineering-2589551
Type: Communication
Title: "Use Of Stromal Intervention and Exogenous Neoantigen Vaccination To Boost Pancreatic Cancer Immunotherapy by Nanocarriers"
The article (in the form of communication) elucidates innovative ways for further improving chemo-immunotherapy using nanotechnology for co-delivery of bioactive agents.
In the Abstract, please rephrase the sentence:
"This is achieved by lipid bilayer coated nanocarriers for achieving synergistic outcomes." in order to avoid ambiguity for readers.
Only one of the 77 References used is from the current year. Try to incorporate more recent citations.
Most of the Figures have no declaration regarding their source or copyright.
Please make it clear the source of the data presented in Figure 2, for instance, (as well as other data presented in other places of the manuscript) and add Reference(s).
In some instances punctuation errors are present, please amend. Examples include: "... single-digit range[1]." (space required after "range"); also in Line 105: "... cycle[10] ...".
In Figure 1 legend please rephrase the sentence: "... inducing chemo agents (e.g., irinotecan, doxorubicin, mitoxantrone) into liposomes ...".
Not much mentioned about the extremely important topic of the adjuvant properties of liposomes. Authors are encouraged to consult following recent publications:
Liposome, nanoliposome and allied technologies in COVID-19 vaccines: key roles and functionalities.
and:
Methodical design of viral vaccines based on avant-Garde nanocarriers: A multi-domain narrative review.
Author Response
This communication paper delves into the utilization of stromal intervention and exogenous neoantigen vaccination for enhancing pancreatic cancer immunotherapy. This paper offers a comprehensive overview of the subject matter and remains highly relevant. However, there are a few minor suggestions for further improvement:
Response: We thank the reviewer for conducting a diligent review and laudatory comments. Our responses to the minor suggestions are as follows:
- Title: Consider changing "chemo-immunotherapy" to "immunotherapy" in the title
to enhance clarity.
R: Thank you for the suggestion. We have changed the title to include “chemo-immunotherapy”
- Introduction: The introduction section appears somewhat brief. Expanding on the application of nanomaterials in pancreatic cancer immunotherapy could provide valuable context.
R: The “introduction” should in reality be seen as an Introductory Statement, followed by two passages (“Nano-enabled chemo-immunotherapy in PDAC; Development of a dual-delivery irinotecan/lipoxin A4 liposome to target the stroma and improve PDAC immunotherapy”) that constitute the introduction. To avoid confusion, we have changed the term “Introduction” to “Introductory Statement”. The following two sections following the introductory statement covers pancreatic cancer immunotherapy in considerable detail, sort of providing a review.
- Introduction, L43-46: Including global statistics rather than focusing solely on the
United States would enhance the paper's international relevance.
R: Thank you, we have supplemented that section by also including global statistics into the statement (L43-45), which now reads:
“Pancreatic ductal adenocarcinoma (PDAC) represents a significant challenge in the field of oncology, ranking as the twelfth most common cancer worldwide with 495,000 new cases in 2020.”
- Figure 1: Could you clarify why bold fonts are used in the caption for L89-90 of the
figure?
- Thank you. We have placed the title for each figure in bold throughout the manuscript, including Figure 1.
- L216-217: If this represents the title of a section, it should be numbered
accordingly. The same applies to L327-328.
R: Thank you. We have numbered all sections and subsections.
- L270: In the figure caption, what does "Fig. S1C" refer to? Clarification is needed.
R: Thank you. Fig. S1C refers to supplementary Figure 1C and it has been revised in the manuscript.
- L367-368: It appears that the figure caption overlaps with the main text of the
paper. Please ensure that figure captions use smaller fonts than the main text to
avoid this issue.
R: Thank you, we have modified the font sizes for all figure legends in the manuscript.
- Including a section on future prospects related to this study would be valuable.
R: Thank you, we have included a description of future prospects into the amended “conclusion” section, change to read Conclusion and Future Prospects. The future prospects discussion (L556-578) read as follows:
“A major future prospect will be to combine nano-enabled chemo-immunotherapy with exogenous vaccination, provided nanocarriers that deliver PDAC-specific neoantigens and tumor-associated antigens (TAAs). We hypothesize that this will lead to strengthening of the cancer immunity cycle, as depicted in Figure 1C. The exogenous vaccination strategy could benefit from a variety of additional smart-design features, including selection of multiple mutant KRAS epitopes that can be assembled into a single vaccine to cover the heterogeneous tumor landscape that may include more than one KRAS mutant. This vaccine will also be valuable for treatment of cancers, other than PDAC. Moreover, neoantigen epitopes can be combined with TAAs (e.g., mucin 1, mesothelin) that, although less immunogenic, are more stably expressed. A multi-epitope strategy will increase the number of antigen specific T-cell clones that can be recruited from adjacent lymphoid structures (including TLS) to participate in the more general immune response triggered by ICD-inducing chemotherapeutic agents. One of the approaches for facilitating multi-epitope delivery include the use of nucleic acid analogs that can be linked together into a single mRNA strand that can be delivered by cationic lipid nanoparticles. These epitope-delivering nanocarriers can also be endowed with surface ligands that target lymphoid structures such as the mannose receptor, widely expressed in antigen presenting cells in lymph nodes and the spleen. In addition, the particles could be endowed with a TLR7 agonist that also serves to strengthen antigen presentation and immune recruitment in lymphoid tissue, which possibly could be extended to TLS developing at the primary cancer site. Thus, an array of vaccination carriers can be developed for use in combination with the liposomes and silicasomes that deliver chemo-immunotherapy agents plus a number of co-packaged immunomodulators, as described by us”
Reviewer 4 Report
This communication paper delves into the utilization of stromal intervention and exogenous neoantigen vaccination for enhancing pancreatic cancer immunotherapy. This paper offers a comprehensive overview of the subject matter and remains highly relevant. However, there are a few minor suggestions for further improvement:
- Title: Consider changing "chemo-immunotherapy" to "immunotherapy" in the title to enhance clarity.
- Introduction: The introduction section appears somewhat brief. Expanding on the application of nanomaterials in pancreatic cancer immunotherapy could provide valuable context.
- Introduction, L43-46: Including global statistics rather than focusing solely on the United States would enhance the paper's international relevance.
- Figure 1: Could you clarify why bold fonts are used in the caption for L89-90 of the figure?
- L216-217: If this represents the title of a section, it should be numbered accordingly. The same applies to L327-328.
- L270: In the figure caption, what does "Fig. S1C" refer to? Clarification is needed.
- L367-368: It appears that the figure caption overlaps with the main text of the paper. Please ensure that figure captions use smaller fonts than the main text to avoid this issue.
- Including a section on future prospects related to this study would be valuable. This could help readers understand potential developments and applications in the field.
Author Response
The article (in the form of communication) elucidates innovative ways for further improving chemo-immunotherapy using nanotechnology for co-delivery of bioactive agents.
- Thank you for your diligent review and making suggestions to improve our manuscript: in accordance with your suggestion, we have made the following edits:
- In the Abstract, please rephrase the sentence:
"This is achieved by lipid bilayer coated nanocarriers for achieving synergistic outcomes." in order to avoid ambiguity for readers.
- Thank you. Based on your suggestion, we have changed the sentence you referred to in the abstract (L23-25), as well as amending the sentence prior to that for cohesiveness. This passage in the abstract now reads as follows:
“We are currently conducting research aimed at enhancing chemotherapy to stimulate anti-tumor immunity by inducing immunogenic cell death (ICD). This is accomplished through the use of lipid bilayer-coated nanocarriers, which enable the attainment of synergistic results.”.
- Only one of the 77 References used is from the current year. Try to incorporate more recent citations.
- Thank you. We have been asked to add a concluding section that also includes future prospects, in which we have included 2023 references. (Ref 3 and 82)
- Most of the Figures have no declaration regarding their source or copyright. Please make it clear the source of the data presented in Figure 2, for instance, (as well as other data presented in other places of the manuscript) and add Reference(s).
- Thank you. We have obtained copyright permission for use of the cancer immunity cycle Figure 1C, and acknowledge that in the figure legend (L104-109).
Figure 2 showing graphical illustration of thin film LXA4 liposomal synthesis and all subsequent figure graphics were created by the authors and have not been published elsewhere. Accordingly, there is no copyright issues
- In some instances, punctuation errors are present, please amend. Examples include: "... single-digit range[1]." (space required after "range"); also in Line 105: "... cycle10] ...".
- Thank you for pointing that out. We have made the following changes:
- L142-145: This design allowed boosting of the irinotecan-induced ICD response in an orthotopic PDAC model through the ability of TLR7 to enhance dendritic cell activation and T-cell recruitment to the PDAC site. (double space)
- L176-181: We are particularly interested in the role CAFs play in programming the TME by a range of soluble mediators, including transforming growth factor-beta (TGF-β), interleukin 6 (IL-6), matrix metalloproteinases (MMPs), vascular endothelial growth factor (VEGF), hepatocyte growth factor (HGF), stromal cell-derived factor 1 (SDF-1), hypoxia-inducible factor-1 (HIF-1), and tissue inhibitors of metalloproteinases (TIMPs).
- L340-342: Four mutations (G12, G13, Q61, G12D), cover more than 99% of the PDAC mutations, with KRAS G12D contributing up to 45% (Fig. 4).
- L347-348: This has resulted in demonstrating therapeutic responses to mutated KRAS peptides in human PDAC trials. (double space)
- L418-420: 5×104 KPC cells, undergoing ICD, were administered in one flank of the animal to induce a protective immune response, demonstrated by growth failure of live KPC cells, injected in the contralateral flank a few days later.
- In Figure 1 legend please rephrase the sentence: "... inducing chemo agents (e.g., irinotecan, doxorubicin, mitoxantrone) into liposomes ...".
- Thank you. We have changed in this sentence as well as the caption and two following sentences in the legend to be cohesive (L88-94). It now reads as follows:
“Figure 1. Use of lipid bilayer encapsulated nanocarriers for endogenous chemo-immunotherapy, to be boosted by KRAS neoantigen-delivering polymer nanoparticles. (A) Schematic to illustrate combination drug loading into liposomes and silicasomes, including remote loading of chemotherapeutic agents that induce immunogenic cell death (ICD), such as irinotecan, doxorubicin, mitoxantrone. The schematic also explains the use of protonating agents (e.g., ammonium sulfate, citric acid, sucrose octa sulfate) for remote import across the lipid bilayer, also serving as a carrier component for incorporating hydrophobic drugs (e.g., paclitaxel, 3M-052) or lipid-conjugated prodrugs, capable of synergizing with the chemotherapeutic agents.”.
- Not much mentioned about the extremely important topic of the adjuvant properties of liposomes. Authors are encouraged to consult following recent publications: Liposome, nanoliposome and allied technologies in COVID-19 vaccines: key roles and functionalities. Methodical design of viral vaccines based on avant-Garde nanocarriers: A multi-domain narrative review.
- Thank you. While agreeing that adjuvant properties of liposomes and viral vaccines are indeed important topics, we were asked in this Communication to limit the scope and theme of discussion to new research in an ongoing research theme, without making it a review that discusses a wide range of topics due to the imposed space limitations. We will keep these topics in mind in future communications.
Round 2
Reviewer 4 Report
I am satisfied with the modifications from the Authors as per my comments. The quality and presentation of this work are improved.
Author Response
Thank you for your comments. We are happy that the changes made were agreeable.